# DB-EAC and LSTR: DBnet based seal text detection and Lightweight Seal Text Recognition

**Baohua Huang**[1]*, **Aokun Bai**[1], **Yuqiong Wu**[2], **Chanjuan Yang**[1], **Han Sun**[1]

1 School of Computer and Electronic Information, Guangxi University, Nanning, China, 2 Auditing Bureau of Xixiangtang, Nanning City, Nanning, China

* bhhuang66@gxu.edu.cn

**Data Availability Statement:** All relevant data are within the paper and its Supporting Information files.

**Funding:** This work was financially supported by the National Natural Science Foundation of China

## Abstract

Recognition of the key text of the Chinese seal can speed up the approval of documents, and improve the office efficiency of enterprises or government administrative departments. Due to image blurring and occlusion, the accuracy of Chinese seal recognition is low. In addition, the real dataset is very limited. In order to solve these problems, we improve the differentiable binarization detection algorithm (DBnet) to construct a model DB-ECA for text region detection, and propose a model named LSTR (Lightweight Seal Text Recognition) for text recognition. The efficient channel attention module is added to the differentiable binarization network to solve the feature pyramid conflict, and the convolutional layer network structure is improved to delay downsampling for reducing semantic feature loss. LSTR uses a lightweight CNN more suitable for small-sample generalization, and dynamically fuses positional and visual information through a self-attention-based inference layer to predict the label distribution of feature sequences in parallel. The inference layer not only solves the weak discriminative power of CNN in the shallow layer, but also facilitates CTC (Connectionist Temporal Classification) to accurately align the feature region with the target character. Experiments on the homemade dataset in this paper, DB-ECA compared with the other five commonly used detection models, the precision, recall, F-measure are the best effect of 90.29, 85.17, 87.65, respectively. LSTR compared with the other five kinds of recognition models in the last three years, to achieve the highest effect of accuracy 91.29%, and has the advantages of a small number of parameters and fast inference. The experimental results fully prove the innovation and effectiveness of our model.

## 1. Introduction

Seal is used to print on the document to indicate identification or signed stationery, because of its production is simple and distinctive signs, widely used in government, enterprises and other organizations issued documents. Although the production methods and styles are not uniform in different countries, most of them have legal effect and occupy an important position in the administrative office. Accurate extraction of key information of seals can efficiently

(Grant no. 61962005), but the funder had no role in study design, data collection and analysis, decision to publish, or preparation of the manuscript.

**Competing interests:** The authors declare that they have no known competing financial interests or personal relationships that could have appeared to influence the work reported in this paper.

organize and classify documents, save office time, having great application value. With the application of deep learning in the field of recognition, character recognition is also ushering in rapid development [1, 2]. At present, the detection and recognition of regular printing fonts and horizontal documents have already achieved high accuracy, but text recognition for complex and irregular scenes is still very challenging.

Unlike conventional text recognition, seal text scene recognition has the following problems: 1) the text is curved and arranged, the outer ring style is not uniform, the text on the background and the seal text tend to form an occlusion interference, coupled with the inclusion of complex patterns, all of which increase the difficulty of seal recognition. 2) Due to the sensitivity of seal data, the lack of public datasets, there are very few resources of real scene images that can be used for model training, which leads to poor model detection and recognition effect and low application value.

The detection and recognition of seals generally includes preprocessing, text area detection and text content recognition steps. Preprocessing includes filtering and denoising, gray scale processing, edge extraction and other methods to improve the effect of text detection and recognition. Although Chinese seal scene recognition has made great progress in the past few years, most of the research work focuses on image preprocessing, and little attention has been paid to the optimization of text detection and recognition models, and the algorithmic models and datasets are based on scenes with low noise. Due to the existence of blurring and deformation of some seals in practical applications, the above models have low accuracy and weak generalization. The complex operation in preprocessing even increases the workload and difficulty of landing the model in the real scene. Some seal detection and recognition models have been proposed to use off-the-shelf end-to-end ABINet [3], PseNet [4] as the detector, outputting both the position of the text and the recognized content. Despite the high efficiency, the accuracy lacks competitiveness compared to two-stage models.

The current research on Chinese seal scene recognition focuses on the preprocessing operation of color comparison, unfolding and arranging the circular text, i.e., using the seal's unique red attribute to highlight the R channel weights, and brightening and reducing the noise of the text part. Ma et al [5] denoised the seal images through preprocessing and used the RGB model to eliminate the brightness of the light-colored parts, and repaired the parts of it that were crippled due to contamination, and highlighted the red color of the seal, the key part of the seal, for the first time, verified the role of preprocessing in seal recognition; Yao et al. [6] tested the HSI color model is better than RGB, and extracted the SIFT features of the seal to be tested, and searched for the matching seal in the seal library to get the coarse matching results, and then used the RANSAC algorithm to remove the incorrectly matched points in the coarse matching, and adjusted the size and angle of the matching seal to improve the accuracy of the recognition; Cai et al. [7] normalize of color seals, extract the color of the clay to simplify the image and calculate the magnitude spectrum of the FFT changes in the image, and then construct the feature matrix corresponding to the structure of the fitness ring, to improve the detection rate of dealing with the background complexity and multi-noise images; Zhang et al. [8] use diffuse filling algorithms to enhance the features of the seal image, and then according to the differences between the pixel gray values, find the same region to achieve image segmentation to ensure the seal detection accuracy. Xiao et al. [9] use polar coordinate expansion to unify the direction of the seal text, and use Bessel curves to fit the up and down text region to improve the accuracy of seal region detection. In addition, by combining cross-stage feature fusion and attention mechanisms [10, 11], designing lightweight CNN can also improve accuracy for small target detection and recognition.

Seal detection and recognition belongs to the field of natural scene text recognition, which is generally divided into two parts: text region detection and text content recognition. In text

region detection, DRRG [12] proposes novel unified relational inference network graphs for detecting arbitrary shapes, which first generates a text proposal model via a convolutional neural network (CNN), and then bridges the deep relational inference network using a graphical convolutional network (GCN) to divide each text instance into a series of rectangular components. Evaluation with the text proposal model allows the network to be trained end-to-end. FCENet [13] models text instances in the Fourier domain and uses Fourier contour embedding to generate more accurate detection region boxes for arbitrarily shaped text contours. TextMountain [14] divides the center and border of the text into the top and bottom of the mountain, with each pixel detected similar to a path to the top. The model makes full use of the overabundance of relationships in text information to help text instances better find the text center. DBNet [15] focuses on improving segmentation results by incorporating the process of differentiable binarization into the training period, and the optimized segmentation network can adaptively set the binarization. The optimized segmentation network can adaptively set the binarization threshold, which not only simplifies the post-processing process, but also improves the accuracy and inference speed of text region detection. TCM [16] uses the CLIP model directly for text detection without a pre-training process. It improves the ability of existing methods to train with fewer samples and significantly improves the performance of baseline methods.

Methodologically, scene text recognition can be viewed as a cross-modal mapping from images to character sequences. Usually the recognition algorithm consists of two modules, a visual module for feature extraction and a sequence module for text transcription, e.g., the CRNN [17] model uses CNN to extract visual features, and then feeds into a recurrent layer BiLSTM to extract sequence features, and finally obtains prediction results by CTC loss modeling, which can handle sequences of arbitrary length. As well as GTC [18] adds an attention guiding mechanism to CTC to better learn character alignment and feature representation, and achieves accurate prediction for both regular and irregular scene text while maintaining fast inference speed. The advantages of this type of algorithm are high accuracy and simple model, which are still preferred by some commercial recognizers. However, the contextual semantic relevance is weak, and the performance is poor for fuzzy, curved, and irregular text, such as deformation, occlusion, and other situations that can limit its effectiveness.

Encoder-decoder based approaches became popular with the introduction of Transformer into the vision domain by the VIT model [19]. NRTR [20] proposes non-recursive end-to-end text recognizer that relies solely on the self-attention mechanism for extracting image features and modeling sequences, dispensing with recursion and convolution, and can be trained with more parallelization and lower complexity. Since scene images vary greatly in text and background, a modal transformation block is further designed to efficiently convert 2D input images to 1D sequences and combined with an encoder to extract more discriminative features. SRN [21] proposes mining semantic information to aid text recognition, introducing a global semantic reasoning module to take into account global semantic contextual information, which is more robust compared to the unidirectional serial semantic transfer approach and more Efficient. Global semantic context propagation is captured through multiple parallel paths, which combines visual and semantic information more effectively. MGP-STR [22] builds a conceptually simple but powerful visual STR model that is constructed based on VIT and includes both purely visual models and language enhancement methods. And further, a multi-granularity prediction strategy is proposed to improve the model performance by implicitly injecting the information of linguistic modalities into the model. SVTR [23] uses a single visual model to dispense with sequence modeling, firstly decomposing the image text into small chunks called character components, and designing hybrid global and local chunks to perceive inter and intra-character coarse-grained features. Enabling characters to be

recognized by simple linear prediction is competitive in terms of inference speed and accuracy. DeepSolo [24] introduces a text matching criterion to provide more accurate supervised signals, allowing a single decoder with Explicit Points Solo to simultaneously perform text detection and recognition for more efficient training. The encoder-decoder based model will have better accuracy because of the consideration of contextual information, but slower inference due to character-by-character transcription. And the model essentially relies on stacked self-attention to learn character-associated features for recognition, lacks the inductive bias of convolution, and requires more data-intensive learning information than the LSTR model. However, due to the confidentiality of seals, there is a lack of a large number of public datasets, so it is difficult to train an effective recognition model for seal text scene recognition.

To solve the above problems, this paper uses a two-stage model for seal text detection and recognition. The use of segmentation-based microscopically binarizable text detection algorithm is naturally suitable for curved text, and competitive results can be achieved without complex preprocessing. And we improve the convolutional recurrent neural network for text recognition of candidate boxes, introduce an inference layer (Inference block) to improve the feature fusion of contextual information, and pay more attention to the correlation between characters. In the real dataset, the high training efficiency and accuracy are still maintained for the cases of font blurring, distortion, and occlusion. First, the seal image is fed into the backbone network Resnet [25] to extract visual features, and the BottleNeck layer in it is improved to postpone downsampling to reduce the loss of image feature information. Second, the feature sequence is fed into the feature pyramid network and added into the Efficient channel attention module [26] (Efficient channel attention), which uses 1*1 convolution cross-channel interaction to further improve the diversity of the feature sequence. The biplot and threshold map are inferred by fusing multi-scale visual features through the feature cascade to determine the detection frame location. Again, the detection frame region is fed into the convolutional layer, and the visual features are extracted and fed into the self-attention-based inference layer. The parallel computation with multi-head attention makes the prediction of sequence labels faster and the learning ability of character granularity features stronger, which can effectively improve the accuracy of the text recognition model. Finally, using CTC transcription, the probability sum of all possible paths for each character is calculated, and a '_' is inserted between repeated characters in the text labels, to determine whether consecutive identical characters are merged. In this way, the label distributions output from the inference layer are sequentially aligned to obtain the final recognition results.

The main contributions of this paper are as follows:

1. Using a model of text detection + text recognition is easier to optimize module by module than an end-to-end approach. Improve the convolutional layer of the differentiable binarization detection algorithm by delaying downsampling to obtain richer semantic features.

2. Introduce an efficient channel attention module into the differentiable binarization detection algorithm, which allows the model to interact across channels, enhances the ability to detect multiscale targets, and speeds up the model convergence rate.

3. A self-attention based inference layer is proposed to construct LSTR to improve the ability to learn multi-granularity character features and reduce the waste of training resources to enhance the model generalization on noisier datasets.

## 2. Text detection and recognition model

### 2.1 Overall structure

Our proposed method as a whole can be divided into two parts, (1) using segmentation-based suitable for curved text of the differentiable binarization algorithm to construct the detection model DB-ECA, and improve it, which can reduce the workload of preprocessing and streamline the model structure (2) Propose a lightweight seal recognition model LSTR, which can achieve high recognition accuracy for fuzzy, deformed and other irregular Chinese seals without collecting a large number of real seal datasets by using CNN's inductive bias and self-attention. The overall structure of this paper is shown in Fig 1.

### 2.2 DB-ECA

Our DB-ECA network structure is shown in Fig 2. First, image visual features are extracted by ResNet and fed to the main stem of the feature pyramid, which is sequentially downsampled in multiples of two to generate feature maps of different sizes. Each layer is then upsampled by a factor of two and fused with the graph of the previous level to merge the deep semantic features and shallow image features, thus capturing multi-scale visual features. Second, the fused multi-scale feature maps are sampled to a quarter of the size of the original image and feature cascading is performed to obtain F. Finally, the probability map P and the threshold map T are predicted using F. The approximate binary mapping B is then computed from P and F. Supervised training is performed on P, T, and B during the training period, and P and B use the same supervised signal (label), and only P or B is needed in the inference phase to obtain the text box.

The backbone of the DB detection algorithm is ResNet, and the original BottleNeck module in the ResNet network uses 1*1 convolution to adjust the number of channels and accelerate convergence via batch norm to improve generalization. Then ReLU solves gradient vanishing and promotes feature information transfer. However, BottleNeck uses 1*1 size convolution kernel and step size s is 2 for downsampling, which results in partial semantic loss. Due to the weight sharing of CNN, it will continue to forwardpropagate the residual feature sequence, which is not conducive to the extraction of richer visual features and leads to the degradation of DBnet performance. Therefore, the downsampling is postponed to the second step of 3*3 convolutional kernel, replacing the 3*3 convolutional kernel step size from 1 to 2, and using the average pooling layer instead of the 1*1 convolutional kernel in shortcut for downsampling. Since the width of the convolution kernel is larger than the step size, it is able to fuse all the information on the feature map during the moving process, and more pixel points can be extracted for backpropagation. The improved BottleNeck structure is shown in Fig 3(A).

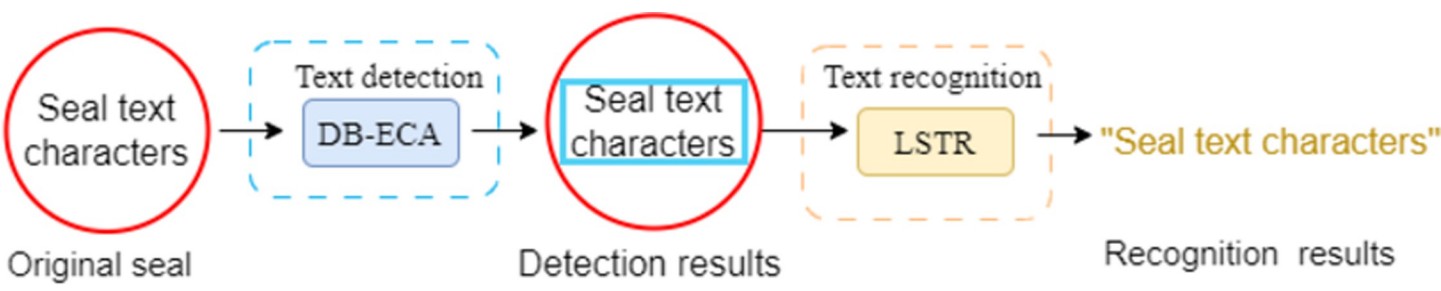

**Fig 1. General structure of the paper.**

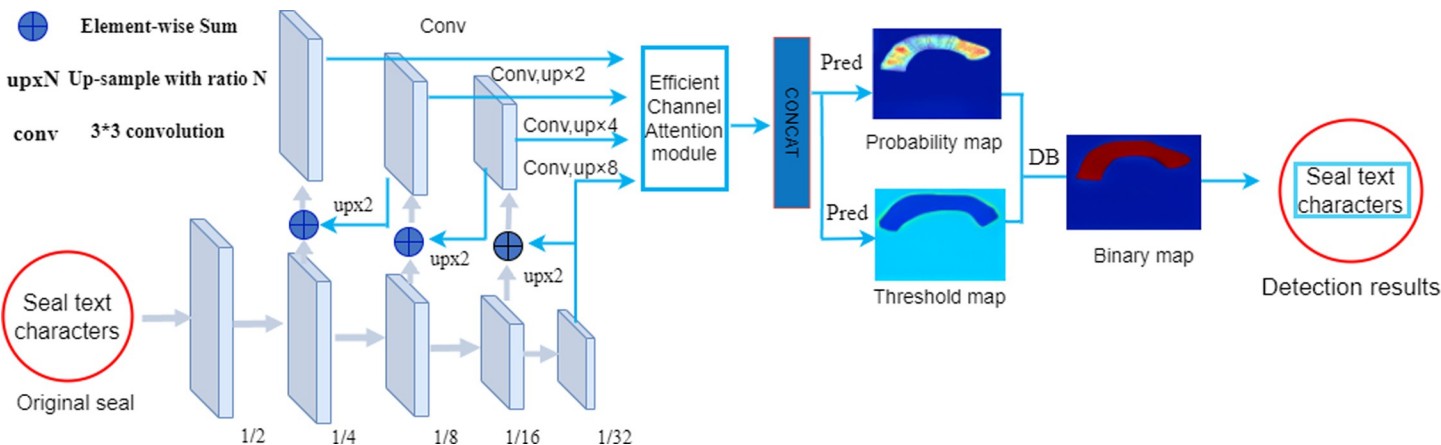

**Fig 2. DB-ECA model diagram.** Where "pred" consists of a 3×3 convolutional operator and two de-convolutional operators with stride 2. The "1/2", "1/4", . . . and "1/32" indicate the scale ratio compared to the input image.

After extracting visual features from the input image, we use 1*1 size convolutional kernel for downsampling, output a feature map scaled by a multiple of 2, and then perform up-sampling feature fusion, and finally unify it into a quarter size of the original image for feature cascading. But suppose a pixel is assigned as a positive value in the downsampling layer, and is regarded as the background in the upsampling cascade operation, it will cause the conflict between different levels of features. And occupies the main part of the feature pyramid (FPN), which interferes with the gradient computation during the training process. To improve the performance of the FPN, an efficient channel attention module, ECA, is introduced to fuse multiscale features before feature cascading, learn the weight coefficients of different channels, and enhance the ability of the feature pyramid to predict multiscale targets. The structure of the ECA network is shown in Fig 3(B).

The ECA model proposes a local cross-channel interaction strategy without dimensionality reduction, which effectively avoids the effect of dimensionality reduction on the learning effect of channel attention, strengthens the correlation of different channels, and automatically learns multi-scale features. First, a feature map with dimension H*W*C is input and spatial feature compression is applied to the feature map. Second, global average pooling GAP is used in the spatial dimension to avoid dimensionality reduction to obtain the aggregated features of the 1*1*C channel map. Again, for the compressed feature map, 1*1 convolution is used to learn each channel and its k neighboring features for local cross-channel interaction. Here, when doing the convolution operation, since the convolution kernel size affects the receptive field, in order to extract different ranges of features for different input feature maps, ECA uses dynamic convolution kernels to avoid manually adjusting k through cross-validation to determine how many neighboring feature matrices are involved in the attentional prediction of a channel. Finally, the aggregated feature map 1*1*C is multiplied with the original input feature map H*W*C to output the feature map with channel attention.

The dynamic convolutional kernel size is changed adaptively by a function, the convolutional kernel adaptive function is defined in Eq (1):

$$k = \psi(C) = |\frac{\log_2(C)}{\gamma} + \frac{b}{\gamma}|_{odd} \tag{1}$$

Where *k* denotes the kernel size, i.e., the coverage of local cross-channel interactions, and how many "neighbors" are involved in the attentional prediction of a channel. c denotes the

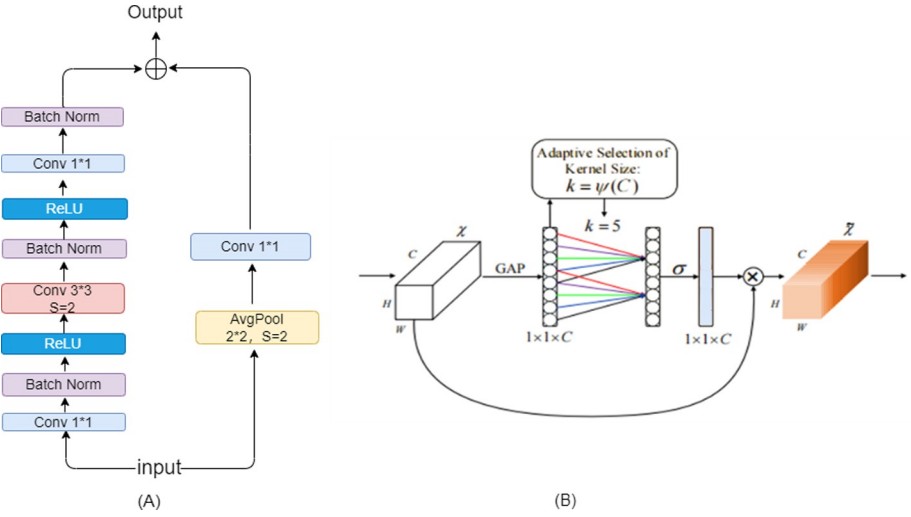

**Fig 3.** (A) BottleNeck structure diagram (B) ECA structure diagram.

number of channels, and *odd* denotes that k can only be taken as an odd number, and γ and *b* are set to 2 and 1, which are used to change the ratio of the number of channels, C, to the size of the convolution kernel and the sum of the kernels. The DB feature pyramid structure with efficient channel attention is shown in Fig 4.

The ECA module is added before feature cascading (CONCAT) is performed. The range of local cross-channel interaction is determined by the kernel size k and efficiently implemented by 1D convolution. So that the model effectively extracts the key character features in the image and reduces the waste of training resources. The experimental results demonstrate that the ECA module captures the cross-channel attention interaction in a lightweight manner, which effectively improves the accuracy and robustness of the detection model.

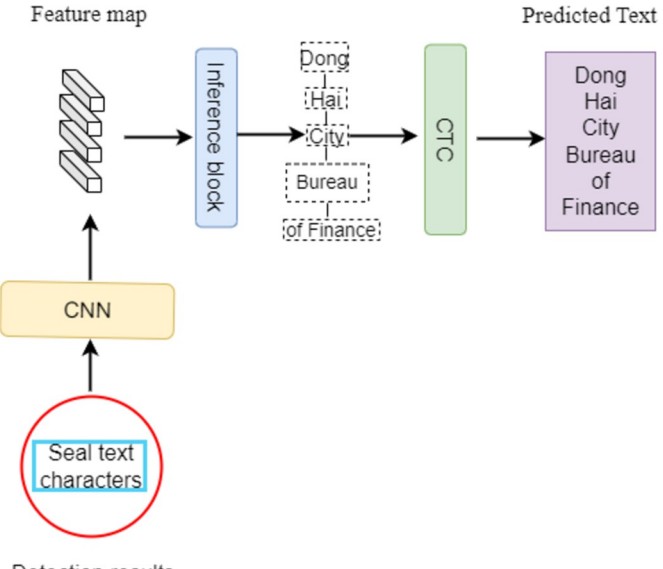

**Fig 4. LSTR structure diagram.**

In computing the approximate biplot by P and F, the standard binarization cannot be back-propagated, much less used directly for training, because it is not differentiable. DBnet proposes to use a differentiable approximate binarization function. The calculation is shown as

$$\widehat{B}_{i,j} = \frac{1}{1 + e^{-k(P_{i,j} - T_{i,j})}} \tag{2}$$

$\widehat{B}$ represents the approximate biplot, and T represents the adaptive threshold map learned from the network. k is the expansion factor, which is empirically set to 50 for the purpose of increasing the gradient and speeding up the convergence. i, j denote the coordinates of the pixel points. The loss function for positive and negative samples is defined as:

$$l_+ = -log\frac{1}{1 + e^{-kx}} \tag{3}$$

$$l_- = -log\left(1 - \frac{1}{1 + e^{-kx}}\right) \tag{4}$$

Find the partial derivative for the input $x$:

$$\frac{\partial l_+}{\partial x} = -kf(x)e^{-kx} \tag{5}$$

$$\frac{\partial l_-}{\partial x} = kf(x) \tag{6}$$

$k$ is the gradient gain factor, through the parameter k can achieve the effect of increasing the gradient magnitude and speeding up the convergence, more favorable optimization, and the final segmentation results will be more superior. The total Loss function is defined as shown in Eq (7)

$$L = L_s + \alpha \times L_b + \beta \times L_t \tag{7}$$

$L_s$ is the Loss of the probability map P and $L_b$ is the Loss of the binary graph B using the binary cross entropy (BCE) operation. $L_t$ represents the threshold graph los, the $\alpha$ and $\beta$ are taken as 1 and 10, respectively. The formula for calculating cross-entropy is given in equation:

$$L_s = L_b = \sum\nolimits_{i \in S_i} y_i \log x_i + (1 - y_i)\log(1 - x_i) \tag{8}$$

where, $S_i$ is the filtered dataset where positive and ne-gative samples are sampled in a ratio of 1:3.The formula used for $L_t$ is shown in Eq (9):

$$L_t = \sum\nolimits_{i \in R_d} |y_i^* - x_i^*| \tag{9}$$

$R_d$ refers to all pixels in the region G obtained by expanding the labeling frame by the D offset, and $y_i^*$ denotesthe labeling of the computed threshold map.

## 2.3 Lightweight Seal Text Recognition model

Lightweight Seal Text Recognition (LSTR) model consists of three parts. Firstly, image visual features are extracted using CNN convolutional layer. Secondly, sequence modeling prediction is performed by inference block. Finally, the recognition results are translated by CTC to transform the feature sequence frames output from the inference layer into labeled sequences. After

experiments, it is proved that LSTR has a smaller number of parameters and faster inference than VIT-based recognition model, and also has advantages in accuracy. The LSTR flow is shown in Fig 4.

VIT as backbone is larger compared to CNN with many parameters model, which will lead to more memory occupation and waste of training resources. And Vit is difficult to obtain the underlying information on small sample datasets, and the number of stacked layers is limited. Therefore, ResNet is used instead as a convolutional layer for visual feature extraction of stamp images.

When the input inference layer image sequence is very long, it will bring a lot of computational time and burden. CRNN solves the problem of RNN gradient explosion or disappearance by introducing LSTM, but the running memory is small, it will limit the cross-sample batch processing, and it will still be tricky to deal with 10N or longer sequences. Transformer-based methods often lack positional encoding, making it difficult to accurately align feature regions with the target object, and are computationally expensive.

In this paper, inspired by the application of Bifomer [27] in vision, a two-layer routing attention mechanism is utilized to filter out most of the irrelevant K-V pairs in the coarse-grained region, followed by applying token-to-token attention to focus on a small portion of the relevant tokens, which provides good performance and computational efficiency since it does not distract the attention of irrelevant tokens. We propose to use stacked multi-head self-attention to construct an inference layer. The number of operations to compute the association between two long-distance text positions does not increase with the distance length, and inter-character visual and positional features are processed in parallel. And the self- Attention [28] mechanism continuously learns intra- and inter-character multi-granularity features to produce a more explanatory model with stronger generalization for irregular text. The inference layer network structure is shown in Fig 5.

Firstly, the visual features are fed into the Multi-Head Attention Multi-Head Attention module to find the association between characters, different weights will be assigned to different strokes and feature extraction will be performed on different scales, which can perceive multi-granularity character features. Secondly, the Dropout function is used to randomly deactivate some neurons to prevent overfitting, and the Layer norm normalization ensures the stability of the data feature distribution and accelerates the convergence speed of the model to solve the gradient vanishing problem. Again, the nonlinear transformation of the input sequence is performed by MLP block to enhance the expression ability of the model, so as to better capture the relationship between the character vectors and improve the performance of the model. Finally, the output results of MLP block are normalized to improve the prediction accuracy of CTC sequence modelling.

The essence of multi-head attention is to unite the learned information from different single-head attention head parts, which can more effectively obtain the correlation between distant characters and filter the character features. The single-head attention formula is shown in Eq (10):

$$Attention(Q, K, V) = Softmax\left(\frac{QK^{\mathrm{T}}}{\sqrt{d_k}}\right)V \tag{10}$$

$Q$ represents the query matrix what is about to be queried, $K$ represents the key matrix i.e. the information being queried and $V$ represents the value matrix i.e. the queried content. First, use the transposed dot product of $Q$ and K to get the degree of correlation of the two matrices, i.e., the degree of similarity of the two feature sequences. Second, divide by $\sqrt{d_k}$ (square root of K dimensions) to keep the gradient stable during training and normalize with Softmax to

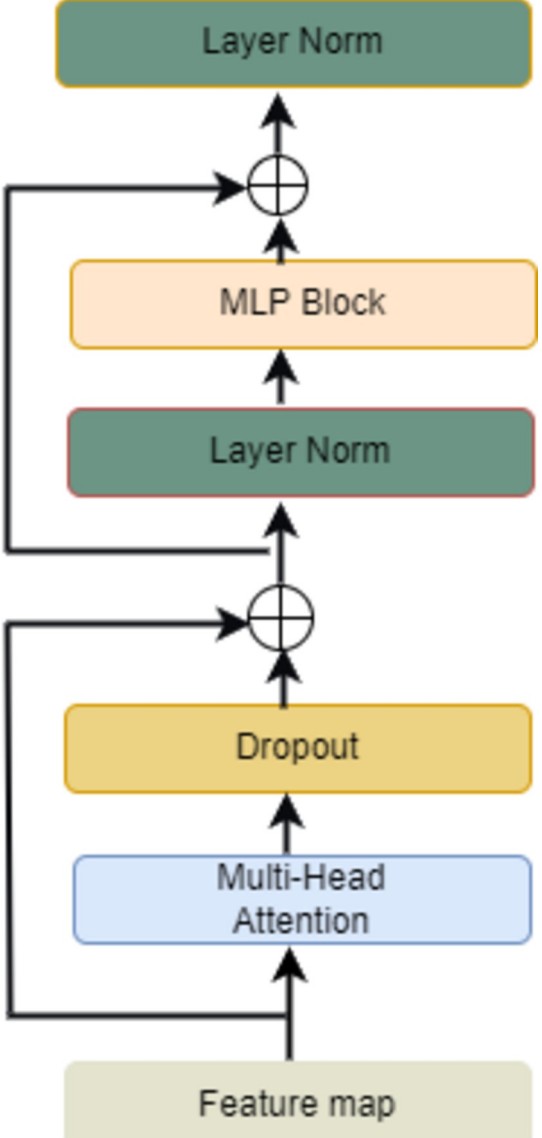

**Fig 5. Inference layer model diagram.**

get a weight matrix. Again, weighting with the content V to make the value matrix more focused on the stroke characteristics of the character, and finally get the feature vector of attention.

Single-head attention has difficulty learning global feature weights and can only establish a Q and K dependency. To capture various ranges of dependencies within the sequence, different stroke features are learned from different subspaces. Multi-head Attention computation is required, where Attention computation is repeated through multiple parallel heads, with different heads processing different information. The multi-head Attention formulas are shown in Eqs (11) and (12):

$$\text{head}_i = Attention\left(QW_i^Q, KW_i^K, VW_i^V\right)W \tag{11}$$

$$MulHead(Q, K, V) = Cancat(\text{head}_1, \cdots, \text{head}_h)W^O \tag{12}$$

$W_i$ denotes the corresponding weight matrix of $Q$, $K$, and $V$. Attention denotes the attention computation, and $\text{head}_i$ represents the computation result of the ith head of attention. $W^o$ denotes the result splicing matrix resulting from the computation of the h heads of attention. In the calculation process, the attention calculation results $\text{head}_i$ of each head are first obtained, and the results are spliced and dot-multiplied with the corresponding weight matrix $W^o$ to obtain the result of multi-head attention.

Finally, the predictions made by the Inference block for each feature vector are converted into a sequence of labels through the CTC loss function. A blank mechanism is introduced to insert a "-" between repeated characters in a text label to solve the problem of whether consecutive identical characters are merged or not, and the CTC loss function is defined as shown in Eq (13):

$$P(l \mid x) = \sum_{\pi \in B^{-1}(l)} P(\pi \mid x) \tag{13}$$

where $B^{-1}(l)$ represents the set of all possible paths that can be merged successfully, and $\pi$ is one of those paths. The probability of each path is the product of the individual time steps and the corresponding character scores. This way of adding up the probabilities of all paths for the matching transformation makes CTC solve the problem of indeterminate-length sequence alignment without the need for an accurate cut of the original character sequence. A maximum likelihood estimation operation on the loss function allows back propagation of the previous neural network, updating the optimizer parameters used to find the most probable character corresponding to the pixel region.

## 3. Experiment

### 3.1 Dataset

The dataset used in this paper includes Chinese seal location detection and text recognition dataset, the text location detection dataset is obtained by the authors of this paper through filming and data enhancement, and the text recognition dataset is constructed by cutting the detected text area box in Python language. The dataset contains 2351 seals, including 676 real seals and 1675 electronic seals (S1 Appendix). Adding analog simulation effects such as rotation and transparency in the production of electronic seals to enhance the training effect and generalization ability of the model for seals with different text orientations. Each sample was uniformly scaled to 420×420 pixels and saved in JPG format. The testing process was divided into training and testing groups with a ratio of 8:2.

### 3.2 Evaluation indicators

ICDAR [29] categorizes text localization into three challenges, challenge 1, 2, and 4, according to the source of the dataset, and each challenge evaluates different methods of detection. In this paper, the data comes from real scene collection and computer generation, and the methods of challenge 1 and 2 are applied, and the accuracy P, recall R, and F measure are used as

the evaluation criteria. The definitions are shown in Eqs (14), (15) and (16):

$$P = \frac{TP}{TP + FP} \tag{14}$$

$$R = \frac{TP}{TP + FN} \tag{15}$$

$$F = \frac{2 \times P \times R}{P + R} \tag{16}$$

*TP* represents the case where the positive sample is predicted to be true, *FP* represents the case where the negative sample is predicted to be true, and *FN* represents the casewhere the true sample is predicted to be false.

## 3.3 Experimental platform

The experiments were done under Windows OS based on Python 3.7, CUDA11.6 Cudnn 8.4.0, CPU i7-9700, GPU Tesla V100 32GB.

## 3.4 Ablation experiments

In the training of the text detection model DBNet-ECA, the iteration number epoch is set to 50, the batch data volume batch-size is 8, the sub-process num_workers is 8, and the initial learning rate is set to 0.001 to reach the minimum value of loss faster. The preprocessing stage applies basic data enhancement techniques such as plus or minus ten degree rotation, cropping, partial flipping and color change, and the processed image is resized to 420*420 to improve the training efficiency. balance_loss is set to true, so that the DBLoss is balanced by default for positive and negative samples in the ratio of 1:3. The threshold of thresh binarization is 0.3, which helps to reduce the situation of misjudging the background region as text. box_thresh text box threshold is set to 0.7, which makes the generation of bounding box more stable and improves the performance of subsequent text recognition. The optimizer is chosen to be Adam. The results for test Precision, recall, and F- Measure are shown in Table 1.

Table 1 compares the precision, recall, and reconciliation averages of ResNet34-DB with ResNet34-DB-ECA and ResNet50-DB-ECA for the three models with or without ResNet improvement, where ResNet50-DB-ECA is the optimal test result, but the number of parameters is also relatively large. The precision rate, recall rate, and reconciliation mean are 90.29%, 85.17%, and 87.65%, respectively. Improving ResNet improves the precision of the three models by 0.85%, 1.08%, and 0.87%, respectively. Without improving ResNet, adding the ECA module makes ResNet34-DB-ECA 2.74%, 4.93%, and 3.92% higher than ResNet34-DB

**Table 1. Detection model comparison.**

| Improving ResNet | Model | Precision% | Recall% | F-Measure |
|---|---|---|---|---|
|  | ResNet34-DB | 81.43 | 74.52 | 77.82 |
| √ | ResNet34-DB | 82.28 | 77.43 | 79.78 |
|  | ResNet34-DB-ECA | 84.17 | 79.45 | 81.74 |
| √ | ResNet34-DB-ECA | 85.25 | 81.79 | 83.48 |
|  | ResNet50-DB-ECA | 89.42 | 82.75 | 85.95 |
| √ | **ResNet50-DB-ECA** | **90.29** | **85.17** | **87.65** |

precision, recall, and reconciliation mean, respectively, and the model size is moderate, and the number of detected frames per second differs from that of ResNet34-DB only by 2.

Since recognizing characters requires a large number of real datasets more, we pre-train LSTR on the public dataset CTW (Chinese Text in the wild) for 100 epochs by transfer learning, and then train it on the homemade dataset. The parameter is set to 200epoch, batch data size of 64, and maximum predicted character length of 40. the recognized character type is set to ch (Chinese) to attenuate numeric or alphabetic interference. The inference layer allows image features to be processed using a cross-attention mechanism, with the number of hidden units being 1024, and the input features are divided into 8 attention heads to capture different positional dependencies. And the dropout deactivation rate is set to 0.1 due to the small real scene dataset. Shuffle is defaulted to true, which ensures that the training images are returned in a different order for each epoch, reducing the overfitting of the model to the training data. fc_decay in CTC is 0.004, allowing the model to learn more detailed features. The Adam optimizer with a weight decay coefficient of 0.05 is used, which requires less memory compared to other optimizers, can adaptively adjust the learning rate of each parameter, and is more suitable for models with large gradient noise in this paper. Data enhancement operations such as rotation, perspective distortion, motion blur, etc. are randomly performed during training.

As can be seen from Table 2. The ResNet34 results are generally better than ResNet18, but the deeper network structure model is also larger. Using ResNet34 as a backbone, the accuracy of the test set is significantly improved by 5.41% and 1.68% compared to VIT and Swin-T. VIT and Swin-T inductive bias is weaker than CNN on small samples, and is poor at reasoning about pixels such as fuzzy irregularities that are not encountered in the model. More data is needed to learn these assumptions automatically, so the accuracy is slightly lower than ResNet34, and the proposed LSTR recognition model combined with the self-attention mechanism has better results for small sample Chinese seal recognition.

Table 3 shows the comparison of the number of parameters and inference speed between Swin-T, VIT-S, ResNet34 and Inference block. The Speed(ms) is the inference time averaged over 100 Chinese seal image text. In order to better highlight the advantages of our model, we use cpu to test inference time (ms). The residual network used in ResNet consists of two 1*1 and one 3*3 convolutions, whereas the token and the hidden size of VIT remain unchanged during the computation process, and the computational complexity is proportional to the token's square, so ResNet has fewer parameters and faster inference speed. Inference block contains only two fully connected layers, the model in this paper not only has the advantage in accuracy, but also has less number of parameters and faster inference speed, which proves the applicability of this method in this study.

### 3.5 Comparison experiments

To further verify the superiority of the detection model DB-ECA and recognition model LSTR in this paper among the existing methods, the comparison experiments of different

**Table 2. Comparison results of different backbone networks.**

| Model | Train accuracy% | Test accuracy% |
|---|---|---|
| ResNet18+BiLSTM | 77.54 | 75.63 |
| ResNet18+inference block | 81.88 | 80.12 |
| VIT+inference block | 85.08 | 84.21 |
| Swin-T+inference block | 88.71 | 87.94 |
| **ResNet34+inference block** | **91.29** | **89.32** |

**Table 3. Backbone network performance comparison.**

| Model | Params(M) | Speed(ms) |
|---|---|---|
| VIT-S+inference | 55.35 | 25.13 |
| Swin-T+inference | 58.21 | 17.27 |
| **ResNet34+inference** | **24.5** | **13.36** |

recognition models are conducted on the homemade real scene dataset in this paper, as shown in Fig 6 and Table 4.

In Fig 6, we can see that TextBoxes++ [30] requires a large number of datasets for model pre-training due to the complex network structure, and the low-level feature expression is weak, CTPN [31] joins the LSTM in the training phase which easily leads to gradient explosion, and it cannot process multi-directional text, so they are not suitable for small-scale target detection such as the seal in this paper, and their precision are respectively lower than those of DB-ECA by 6.71% and 7.63%.Although DRRG can realize the detection of arbitrary shaped text, the detection results are overly dependent on the individual word detection frames suggested by the text component. PRPN [32] proposes a two-dimensional asymptotic kernel to satisfy the requirements of the curved text detection task, but there are difficulties in dealing with text embeddings like seals and other rare training data, and thus the Precision is lower than that of DB-ECA by 4.96%, respectively, 3.48%. The experimental results demonstrate that DB-ECA achieves the highest Precision and F-Measure while maintaining a high FPS.

The transformer-based models NRTR, SRN have larger complexity and parameter counts and require a large number of datasets for gradient updating and optimization, especially for some specific data, so they are not suitable for small-sample studies of real stamp scenes, and their accuracies are lower than that of LSTR by 6.07% and 3.89%, respectively. SVTR unites the extraction of visual and sequence features into a single module, and it is more accurate at the speed is more advantageous, but the accuracy is slightly worse than LSTR 1.81% for stamp images with high noise such as blurring and bending etc. MGP improves the encoder part but lacks a 'localizer' like CTC, and the accuracy is lower than LSTR 1.73%. This proves that LSTR,

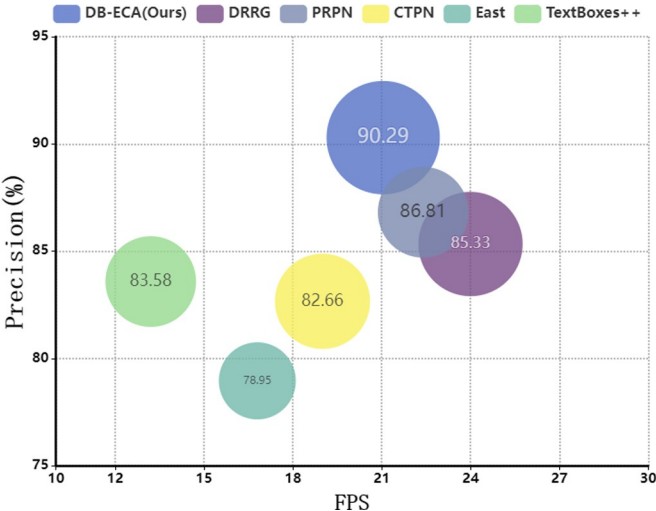

**Fig 6. Detection results of different models.** The number in the center of the bubble indicates Precision, and the area of the bubble is proportional to F-Measure.

**Table 4. Comparison of different recognition models.**

| Model | Accuracy% | Speed(ms) | Params(M) |
|---|---|---|---|
| NRTR | 83.25 | 54.0 | 31.7 |
| SRN | 85.43 | 24.74 | 54.7 |
| SVTR-S | 87.51 | 8.51 | 10.3 |
| ABINet | 86.62 | 15.47 | 36.7 |
| MGP-T | 87.59 | 20.52 | 26.4 |
| **LSTR** | **89.32** | **13.36** | **24.5** |

which adapts to a small sample dataset by CNN and combines CTC to align feature regions with the target object, has advantages in both accuracy and inference speed.

## 3.6 Parameter analysis

In this section, the impact of improving ResNet and adding efficient channel attention on DBnet-based detection models is analyzed experimentally. As shown in Table 1, using average pooling instead of 1*1 convolutional kernel and postponing downsampling can extract richer character semantic features, which all improve the performance of the detection model. The introduction of efficient channel attention and cross-channel interaction before feature cascading significantly improves the accuracy of the detection model and effectively solves the problem of mismatch of weights of the same pixel point on feature maps of different scales. Table 5 calculates the detection frame rate per second of the detection model, which shows that ResNet50 has the largest number of parameters and slower inference speed, while the addition of ECA has less impact on the model speed and the number of parameters. Table 2 compares the recognition models by ablation experiments, due to the introduction of residual network and Batch Normalization in ResNet, the convergence is faster and the number of parameters is smaller, and the interference block outperforms BiLSTM in models with deeper networks. LSTR Compared to the transformer-based recognition model [20, 21], the detection accuracy and inference speed are improved, verifying that the combination of CNN and inference layer can capture long-range dependencies, improve accuracy by making predictions on irregular seal text such as occlusion and blurring based on contextual semantic features, eliminate loops and recursion with parallel computing, and have an advantage in inference speed. The recognition model training accuracy is slightly lower than that of the training set in the test set, indicating that the LSTR recognition model does not suffer from overfitting phenomenon during the migration learning process after the Adam optimizer weight attenuation. It is demonstrated that using the inductive bias of CNN can help the inference layer to speed up the convergence speed and have good global modeling ability on small samples.

## 4. Conclusion

In this paper, DB-ECA seal text location detection model and Lightweight Seal Text Recognition model are proposed. The ECA channel attention is added to solve the DBNet feature

**Table 5. Model performance comparison.**

| Model | Params (M) | FPS |
|---|---|---|
| ResNet34-DB | 20.1 | 34 |
| ResNet34-DB-ECA | 21.9 | 32 |
| ResNet50-DB-ECA | 25.6 | 21 |

pyramid conflict problem, and the inference block is constructed instead of the loop layer. The inference layer utilizes the self-attention mechanism, which can perform global character feature modeling, fusion of contextual semantic features, and parallel prediction of visual features to improve recognition efficiency and accuracy. Experimental evaluation on a self-made real Chinese seal dataset fully verifies that the DB-ECA and LSTR models outperform other models in terms of performance and recognition accuracy, and are able to accurately extract the key information of the seal faster and more accurately. It solves the problem of inefficiency in the traditional administrative office, saves material and financial resources, and is of great help to the government, enterprises and other administrative departments in the intelligent office. Although the recognition method in this paper achieved 91.29% accuracy in the test set, the Chinese seal text recognition has achieved good results. However, the method of CTC loss function decoding has a certain degree of randomness, which will affect the alignment effect of features and labels. Therefore, in the future, we consider adding GCN graph convolutional neural network to the CTC branch of the recognition model to establish the connection between the label and local features, and improve the model expression ability, in order to improve the accuracy of the Chinese seal text recognition model.

## Supporting information

**S1 Appendix. Dataset sources.**
(DOCX)

## Author Contributions

**Conceptualization:** Aokun Bai.

**Data curation:** Yuqiong Wu.

**Funding acquisition:** Baohua Huang.

**Methodology:** Aokun Bai.

**Resources:** Baohua Huang, Yuqiong Wu.

**Supervision:** Baohua Huang.

**Validation:** Aokun Bai, Chanjuan Yang.

**Visualization:** Aokun Bai, Han Sun.

**Writing – original draft:** Aokun Bai.

**Writing – review & editing:** Baohua Huang.

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
