## [Decision Letter · Decision Letter 0]

16 Nov 2023

PONE-D-23-32033DB-EAC and LSTR: DBnet based seal text detection and Lightweight Seal Text RecognitionPLOS ONE

Dear Dr. Huang,

Thank you for submitting your manuscript to PLOS ONE. After careful consideration, we feel that it has merit but does not fully meet PLOS ONE’s publication criteria as it currently stands. Therefore, we invite you to submit a revised version of the manuscript that addresses the points raised during the review process.

We look forward to receiving your revised manuscript.

Kind regards,

Nouman Ali

Academic Editor

PLOS ONE

 [National Natural Science Foundation of China (Grant no. 61962005)].  

6. We note that Figure 1, 4, 5 and 7 in your submission contain copyrighted images. All PLOS content is published under the Creative Commons Attribution License (CC BY 4.0), which means that the manuscript, images, and Supporting Information files will be freely available online, and any third party is permitted to access, download, copy, distribute, and use these materials in any way, even commercially, with proper attribution. For more information, see our copyright guidelines: http://journals.plos.org/plosone/s/licenses-and-copyright.

a. You may seek permission from the original copyright holder of Figure 1, 4, 5 and 7 to publish the content specifically under the CC BY 4.0 license. 

Reviewers' comments:

Reviewer's Responses to Questions

**Comments to the Author**

1. Is the manuscript technically sound, and do the data support the conclusions?

Reviewer #1: Yes

Reviewer #2: Yes

Reviewer #3: Yes

2. Has the statistical analysis been performed appropriately and rigorously? 

Reviewer #1: I Don't Know

Reviewer #2: Yes

Reviewer #3: Yes

3. Have the authors made all data underlying the findings in their manuscript fully available?

Reviewer #1: No

Reviewer #2: Yes

Reviewer #3: No

4. Is the manuscript presented in an intelligible fashion and written in standard English?

Reviewer #1: Yes

Reviewer #2: Yes

Reviewer #3: Yes

5. Review Comments to the Author

Reviewer #1: This article discusses the importance of artificial intelligence in the field of seals, aiming to solve the problem of seal information localization and recognition. Therefore, an efficient DBNet-based model is proposed in this article. The paper has a certain degree of innovation, but there are also serious problems that need to be solved. Recommendations for improving the manuscript:

1) Figure 2 is incompletely plotted, for example, activation functions, normalization operations, and so on. Please plot Figure 2 carefully and add relevant descriptions.

2) Please pay attention to the latex writing specifications for equation 10, such as "radical sign" and "superscript and subscript". Because the symbol "^" is often represented as an XOR operation in computer languages such as Java, C++, and Python.

3) In Section 3.2, the article mentions biformer, an attention mechanism that very cleverly lightweights mhsa through operations such as sparse matrices, lightweight convolution, and so on, without reducing accuracy. Please add the cleverness of biformer and reduce the description of mhsa, because mhsa has become common knowledge in the field of attention mechanisms.

4) Please note the specification of equation 16, which uses "F1" instead of "F". In addition, there are formatting errors in the two lines of text below equation 16, such as font and spacing. Please correct them carefully.

5) In Section 4.4, "The results for training accuracy, recall, and F-value are shown in Table 1." does not match the performance metrics in Table 1, so if English is not the first language, please check after using the translator. The article is written with a terrible attitude. Moreover, why "training accuracy"? Why not use test set results?

6) Why are "Params" in table 2 and table 4 not using the same unit?

7) There may be problems with the two sets of experiments in Table 3. How are "vit+inference block" and "swin-t+inference block" completed? The "inference block" is a variant of mhsa. vit and swin-t are also variants of mhsa. Therefore, how the experiments are completed? Please give a reasonable explanation.

8) Why is the cpu used in Table 4? what does the gpu do in the experimental environment? Please give a reasonable explanation.

9) Why are the performance metrics described differently in each table? This is very unfriendly to reviewers and readers.

10) The performance metrics of the comparative models in Table 5 are too few, so please add the results.

11) It's a well-known fact that the fastest way to get academic resources is not to work hard, but to just loot them. CV field ≈ ctrl c + ctrl v field.

12) The problem of seal information localization and recognition belongs to the problem of small target recognition, but the research on such aspect in these paper are clearly insufficient, for example, the achievement in these two literature have not even been mentioned:DOI:10.1093/jcde/qwac071，DOI:10.1007/s10462-023-10438-y，and so on.

Therefore, please draw your own diagram of the general architecture of the model, not the DBNet prototype, so that the subsequent image structure is represented in the general architecture diagram.

Reviewer #2: 1.The author proposes his own method for the detection and recognition of Chinese seal text, which has practical significance. The overall article is innovative and has sufficient workload.

2.The word detection section can be compared with other algorithms, including classic text detection algorithms based on regression methods. Conduct ablation experiments on the improved part and add more comparative images of experimental results.

3.It should be explained whether seal images with completely opposite text directions can be detected and recognized.

4.The data volume is not large enough, and more seal images can be collected or more dataset images can be obtained through transfer learning and other methods.

Reviewer #3: General Comments:

The topic of this manuscript is to improve the accuracy of Chinese seal recognition. In my opinion, there are two innovative points in this paper. Firstly, a model DB-ECA was constructed to improve the differentiable binarization detection algorithm (DBnet). Secondly, a model named LSTR (Lightweight Seal Text Recognition) was propose for text recognition. In my opinion, The method used in this paper is innovative, but the application area is not very attractive. Therefore, I suggest that the manuscript can be published after major revision. Before acceptance for publication, the paper needs following improvement:

1. Line 10-12. The specific significance of this study should be supplemented in the beginning of the abstract. Which fields or works will this research be most helpful for? What is the important significance of extracting text information from Chinese seals?

2. Line 18-23. If this journal does not limit the word count of abstract, I suggest supplementing the results and conclusions of this study in the abstract.

3. Line 25-29. I suggest that the author add the content of the significance and importance of this study at the beginning of the introduction

4. Line 73. Can this section “2.Related Work” be merged with the introduction? Most studies would analyze the methods and results of previous research in the introduction. If it can be merged into Introduction, I suggest rewrite the content of this section. The introduction should be reduced to 5-6 paragraphs with a summary at the end of each paragraph.

5. Line 146-158. A title should be supplemented before the text of this paragraph？

6. Line 158-160. Is Figure 1 original by the author? Can the first input small picture be replaced with the pictures related to this study? I suggest that the title of this figure be written in more detail. It is necessary to add descriptions of important parts and some small figures.

7. I strongly suggest that the author can supplement a diagram as Figure 1 to introduce the overall process of this study.

8. I suggest merging Figures 2 and 3 into one figure and displaying them separately with panel (A) and (B). In addition, the styles and colors of the two small figures should be consistent.

9. Are the three text recognition results in Figure 7 are screenshots of a programming tool? I don't like the style of this figure. Readers outside China cannot understand its meaning. I suggest that the Figure 7 should be redraw. The small pictures of the 3 seals and their text recognition results should be divided into 3 different groups and panels. The figure caption should describe the three panels in details.

10. Line 145-301. I suggest reducing at least 30 lines text of “Text detection and recognition model”. I think the description in this section is not concise, and some redundant content should be reduced.

11. The paper is generally understandable. It is suggested the author check carefully the English writing and use standard terminologies in the neural network areas.

6. PLOS authors have the option to publish the peer review history of their article (what does this mean?). If published, this will include your full peer review and any attached files.

Reviewer #1: No

Reviewer #2: No

Reviewer #3: No

---

## [Author Response · Author response to Decision Letter 0]

19 Feb 2024

Thanks to the comments and suggestions from all reviewers, we revised the paper and described our efforts in the 'Response to Reviewers'.

---

## [Decision Letter · Decision Letter 1]

25 Mar 2024

DB-EAC and LSTR: DBnet based seal text detection and Lightweight Seal Text Recognition

PONE-D-23-32033R1

Dear Dr. Huang,

We’re pleased to inform you that your manuscript has been judged scientifically suitable for publication and will be formally accepted for publication once it meets all outstanding technical requirements.

Kind regards,

Nouman Ali

Academic Editor

PLOS ONE

Additional Editor Comments (optional):

Reviewers' comments:

Reviewer's Responses to Questions

**Comments to the Author**

1. If the authors have adequately addressed your comments raised in a previous round of review and you feel that this manuscript is now acceptable for publication, you may indicate that here to bypass the “Comments to the Author” section, enter your conflict of interest statement in the “Confidential to Editor” section, and submit your "Accept" recommendation.

Reviewer #1: All comments have been addressed

Reviewer #3: All comments have been addressed

2. Is the manuscript technically sound, and do the data support the conclusions?

Reviewer #1: Yes

Reviewer #3: Yes

3. Has the statistical analysis been performed appropriately and rigorously? 

Reviewer #1: Yes

Reviewer #3: Yes

4. Have the authors made all data underlying the findings in their manuscript fully available?

Reviewer #1: Yes

Reviewer #3: Yes

5. Is the manuscript presented in an intelligible fashion and written in standard English?

Reviewer #1: Yes

Reviewer #3: Yes

6. Review Comments to the Author

Reviewer #1: The authors have revisited manuscript carefully. And the theory is reasonable and has application value. Therefore, there are no more comments, it could be accepted at present form.

Reviewer #3: (No Response)

7. PLOS authors have the option to publish the peer review history of their article (what does this mean?). If published, this will include your full peer review and any attached files.

Reviewer #1: No

Reviewer #3: No

---

## [Editor Report · Acceptance letter]

2 May 2024

PONE-D-23-32033R1 

PLOS ONE

Dear Dr. Huang, 

I'm pleased to inform you that your manuscript has been deemed suitable for publication in PLOS ONE. Congratulations! Your manuscript is now being handed over to our production team.

Kind regards, 

on behalf of

Dr. Nouman Ali 

Academic Editor

PLOS ONE